# MATPOOL: MATRIX-PATTERN-ORIENTED POOLING FOR GRAPH PROPERTY PREDICTION

## ABSTRACT

Graph property prediction usually involves using a model to predict the label for the entire graph, which often has complex structures. Because input graphs have different sizes, current methods generally use graph pooling to coarsen them into a graph-level representation with a unified vector pattern. However, this coarsening process can lead to a significant loss of graph information. In this work, we explore the graph representation by using a matrix pattern and introduce an algorithm called Matrix-pattern-oriented Pooling (MatPool) that provides a unified graph-level representation for different graphs. MatPool multiplies the transposed feature matrix by the feature matrix itself and then conducts an isomorphic mapping to create a Matrix Representation (MR) that preserves the graph information and satisfies permutation invariance. Since the multiplication operation calculates the relationships between each feature, MR exhibits row-column correlations under the matrix pattern. To match this correlation, MatPool uses a novel and efficient Matrix Neural Network (MNN) with two-sided weight matrices to match the row-column correlation under the matrix pattern. We provide theoretical analyses to reveal the properties of MatPool and explain why it can preserve graph information and satisfy the permutation invariance. Extensive experiments on various graph property prediction benchmarks show the efficiency and effectiveness of MatPool.

## 1 INTRODUCTION

Graph-structured data Hu et al. (2020) are everywhere and play a key role in social networks Fan et al. (2022), recommender systems Liu et al. (2023b), transportation Ye et al. (2020), and protein prediction Gao et al. (2023). With the development of Graph Neural Networks (GNNs) Wu et al. (2021; 2022b), GNNs are excellent at handling tasks that predict properties for individual nodes Gao et al. (2018); Wu et al. (2023). When working with individual nodes, the goal is to predict labels based on their connections and features within the graph.

In contrast, graph property prediction Xie et al. (2022) predicts labels for entire graphs, which can vary significantly in size. Machine learning algorithms Bishop (2007); Goodfellow et al. (2016) usually require data in a unified size, but the different sizes of graphs make it hard to use them directly as inputs. Therefore, developing methods to provide a uniform graph-level representation is crucial for accurately predicting graph properties and improving the effectiveness of GNNs in various graph-level applications.

Similar to how pooling in Convolutional Neural Networks (CNNs) Rawat & Wang (2017) uses down-sampling to gather feature information, graph pooling Liu et al. (2023a); Jin et al. (2020) generally applies down-sampling to coarsen node information into a unified graph-level representation with a consistent vector size. This approach is especially useful for graphs that change in size, as it makes it easier to apply these representations to various machine-learning techniques.

**Motivation**: However, downsizing and coarsening in pooling methods often lead to significant loss of graph information. To address this issue, we explore a new way of representing graphs using a matrix pattern that retains graph information. Additionally, we design a specialized neural network to handle this matrix pattern, capturing the relationships between its rows and columns.

**Why matrix?** The features of entire graphs are presented in the form of matrices. Although the number of nodes in different graphs varies, resulting in different feature scales, all these graphs can be transformed or mapped into a unified matrix space through isomorphic mappings. This means that, despite the potential differences in their original features, through such transformations, they can be compared, analyzed, or processed within a common space. Moreover, matrix pattern can provide a more detailed and richer view of graphs than vector pattern.

Consequently, we propose Matrix-pattern-oriented Pooling (MatPool) for graph property prediction. MatPool has two key components: Matrix Representation (MR) to handle graphs of different sizes and Matrix Neural Network (MNN) to explore the features of MR more deeply.

**Matrix Representation (MR)**: We design a message-passing way called Positive Eigenvalue Mapping (PEM), enhancing the propagation influence of primary nodes and making the eigenvalues of the adjacency matrix positive. Next, we multiply the transposed feature matrix by the feature matrix itself and perform a isomorphic mapping to create a Matrix Representation (MR) for each graph.

**Matrix Neural Network (MNN)**: The matrix-pattern-oriented MR for each graph exhibits strong connections between rows and columns, which traditional CNNs or MLPs struggle to capture. To address this, we design a Matrix Neural Network (MNN) that uses two-sided weight matrices, allowing for more effective output calculations and naturally captures the row-column correlations.

As a result, we propose MatPool, a graph-level learning framework designed to predict properties for graphs of varying sizes without losing graph information. The contributions of this study are summarized as follows:

- We create a Positive Eigenvalue Mapping (PEM), which enhances the propagation influence of primary nodes, to aggregate the node features in the graph.

- We propose a Matrix Representation (MR) that isomorphically maps the varying graph feature space into a unified graph-level space without lossing graph information.

- We design a novel Matrix Neural Network (MNN) that uses two-sided weight matrices to efficiently capture row-column correlations and extract deeper features from the MR.

- We provide theoretical analyses to reveal the properties of MatPool and explain why it can preserve graph information and satisfy permutation invariance. Extensive experiments validate the efficiency and effectiveness of MatPool.

The rest of this paper is organized as follows: Section 2 briefly surveys related work on graph representation learning and matrix learning. Section 3 describes the detailed implementation of MatPool. In Section 4, experimental results on benchmark graph datasets demonstrate the effectiveness and efficiency of MatPool. Finally, the conclusion is presented in Section 5.

## 2 RELATED WORK

**Graph Neural Networks (GNNs)**: GNNs are powerful machine learning algorithms for processing graph-structured data. They capture the dependency relationships between nodes during the message-passing process, enabling accurate and comprehensive analysis and prediction. As a novel learning technology, GNNs continue to attract significant research interest and find applications in various fields Gilmer et al. (2017); He & Zhao (2020); Saha et al. (2022). The Graph Convolutional Network (GCN) Kipf & Welling (2017) is the most typical message-passing method for graph data, aggregating node information from downstream layers.

With in-depth research on graph-structured data, many algorithms have been proposed in recent years to address various problems such as heterogeneous graphs Wang et al. (2019); Ma et al. (2022), over-smoothing Keriven (2022); Wei et al. (2023), and more Chen et al. (2022); Zhu et al. (2023). For example, GraphSAGE Hamilton et al. (2017) utilizes sampling technology to solve non-inductive and non-batch training issues. GIN Xu et al. (2019) improves the performance and efficiency of graph neural networks based on the WL-test Shervashidze et al. (2011) that analyzes the expressive ability of GNNs for different graph structures. DeepGCN Li et al. (2019; 2021b) and DeeperGCN Li et al. (2021a) draw inspiration from the residual idea, modifying the propagating and aggregating framework to adapt to the training of deep models.

**Graph Pooling**: The size of different graphs often varies, making it difficult for algorithms to handle these size-varying graphs directly. Therefore, a unified graph-level representation is essential for graph property prediction. Graph pooling methods Shen et al. (2018); Lee et al. (2018); Wang & Ji (2023) effectively provide this unified representation for graphs of different scales. These methods can be categorized into global pooling and hierarchical pooling.

Global pooling methods Yuan & Ji (2020a); Bianchi et al. (2020b) consider the information of all nodes and pool the entire graph directly. For example, Set2Set Vinyals et al. (2016) finds the importance of nodes to provide a unified graph-level representation for different graphs. Global-Attention Li et al. (2016) uses an attention mechanism to aggregate entire graph information. Sort-Pool Zhang et al. (2018) transforms the nodes by sorting and concatenating them. However, global pooling may overly compress graph information during downsizing and coarsening.

Hierarchical pooling methods Gao et al. (2022); Wu et al. (2022a) aggregate node features in a hierarchical structure. For example, DiffPool Ying et al. (2018) uses a differentiable pooling layer to form a fixed number of clusters. TopK Gao & Ji (2019) scores nodes using a trainable projection vector and samples them based on their scores. Self-Attention Pooling Lee et al. (2019) improves TopK by attention scores. Adaptive Structure Aware Pooling Ranjan et al. (2020) uses a self-attention network to learn graph information by hierarchically capturing local subgraph information. These hierarchical pooling methods align the size of graphs during the coarsening process.

**Matrix Learning**: Matrix and vector features are two common data representations in machine learning. Compared to vector features, matrix features Wang et al. (2008) provide a more detailed and richer view, which is crucial for representing graph-structured data. Matrix features also naturally express the interaction between features. Early methods like MatMHKS Chen et al. (2007) were designed to handle matrix samples, such as images, without converting them to vectors, preserving the spatial structure. Algorithms like MLMMPC Zhu et al. (2015) and BPDMatMHKS Wang & Zhu (2018) integrate localization information to improve performance. More recent methods, such as EMatMHKS Zhu et al. (2020), greatly accelerate the training speed of MatMHKS and demonstrate the generalization ability of matrix classifiers.

Although these matrix classifiers can be directly applied to matrix samples, they optimize the objective function using the Moore-Penrose inverse under the minimum square error. Consequently, they fail to form a deep learning framework.

**Relations to Our Work**: Key differences between our work and related research can be summarized as follows: (i) We create a propagating and aggregating way called Positive Eigenvalue Mapping (PEM), which differs from existing GNNs; (ii) We provide a graph-level Matrix Representation (MR) and reveal its properties, which differ from existing vector-based models; (iii) We design a novel Matrix Neural Network (MNN) using two-sided weight matrices to extract deeper features from the MR.

## 3 PROPOSED METHOD

### 3.1 PRELIMINARIES

Let $\mathcal{G}(\mathcal{X}, \mathcal{E})$ denote a directed or undirected graph with node set $\mathcal{X} = \{x_1, x_2, ..., x_n\}$ and edge set $\mathcal{E} = \{e_{11}, ..., e_{i_1 1}, e_{12}, ... e_{i_2 2}, ..., e_{1n}, ..., e_{i_n n}\}$, where $x_i \in \mathbb{R}^{d_{\mathcal{X}}}$ stands for the feature of the $i^{th}$ node and $e_{ji} \in R^{d_{\mathcal{E}}}$ as the feature of the edge connecting node $x_j$ to node $x_i$.

To simplify the process and proof, we will use a matrix approach and redefine $\mathcal{G}(\mathcal{X}, \mathcal{E})$ as $\mathcal{G}(A, X, E)$. Here, $A \in \mathbb{R}^{n \times n}$ denotes the adjacency matrix without self-loop and $A_{i,j} = 1$ stands for the $j^{th}$ node $x_j$ is connect to the $i^{th}$ node $x_i$, $X = [x_1, x_2, ..., x_n]^T \in \mathbb{R}^{n \times d_{\mathcal{X}}}$ denotes the feature matrix of nodes. $E = [e_1, e_2, ..., e_n]^T \in \mathbb{R}^{n \times d_{\mathcal{E}}}$ denotes the edge feature matrix, where $e_i = \sum_{x_i \in \mathcal{N}(x_j)} e_{ji}$ represents the sum of the edge features directed towards $x_i$.

Given a set of graphs $\{\mathcal{G}_1, \mathcal{G}_2, ..., \mathcal{G}_N\}$, where the graphs have varying sizes of nodes, the primary goal of the graph pooling function $Pool$ is to provide a unified graph-level representation for each graph. Suppose the function $Size$ returns the shape of the matrix. Then, the goal of $Pool$ can be described as follows,

$$Size(Pool(\mathcal{G}_i)) = Size(Pool(\mathcal{G}_j)), \forall i, j \leq N \tag{1}$$

Table 1: Comparison of message-passing and aggregating process schemes.

| MODEL | MESSAGE-PASSING AND AGGREGATING |
|-------|----------------------------------|
| GCN | $\tilde{D}^{-0.5}(A^T + I)\tilde{D}^{-0.5}X + E'$ |
| GIN | $(A^T + (1+\epsilon)I_n)X + E$ |
| PEM | $A_{pem}X + E$ |

### 3.2 POSITIVE EIGENVALUE MAPPING

We propose a message-passing way named Positive Eigenvalue Mapping (PEM) that keeps the eigenvalues of the adjacency matrix positive and enhances the propagation influence of primary nodes. PEM is the groundwork for subsequent graph-level representation.

Firstly, considering both directed and undirected graphs, we calculate the normalized adjacency matrix as follows,

$$\hat{A} = (a_{ij})_{n \times n} = D^{-0.5}A^T D^{-0.5} \tag{2}$$

where $D$ is a diagonal matrix and each diagonal element $D_{i,i} = \sum_{j=1}^{n} A_{j,i}$. Next, we calculate the reconstructed adjacency matrix as follows,

$$A_{pem} = \begin{bmatrix} \epsilon + \sum_{i=1}^{n} a_{1i} & \dots & a_{1n} \\ \vdots & \ddots & \vdots \\ a_{n1} & \dots & \epsilon + \sum_{i=1}^{n} a_{ni} \end{bmatrix} \tag{3}$$

where $\epsilon$ is a small positive perturbation value.

**Proposition 3.1.** *Given an adjacency matrix $A \in \mathbb{R}^{n \times n}$, let $\lambda_i$ be the eigenvalue of the reconstructed adjacency matrix $A_{pem}$. Then, $\forall i \leq n, \lambda_i > 0$.*

Suppose the dimensions of the node and edge features are equal. The process of propagating and aggregating in PEM, without considering the neural network mapping, is conducted as follows:

$$Agg(\mathcal{G}(A_{pem}, X, E)) = A_{pem}X + E \tag{4}$$

Table 1 lists the message-passing processes of various GNNs, including GCN, GIN, and PEM. From the table, it is evident that PEM maintains relatively high diagonal values in $A_{pem}$, indicating that PEM enhances the propagation influence of primary nodes.

In GCN, the re-normalization process is used, where $\tilde{D}$ is a diagonal matrix with each diagonal element $\tilde{D}_{i,i} = 1 + \sum_{j=1}^{n} A_{j,i}$. GCN considers the weight of edges, and each $e'_i$ in $E'$ can be calculated as $e'_i = \sum_{x_j \in \mathcal{N}(x_i)} \tilde{D}_{i,i}^{-0.5} \tilde{D}_{j,j}^{-0.5} e_{ji}$. Moreover, GIN treats all nodes roughly equally.

### 3.3 MATRIX REPRESENTATION

We design a graph-level Matrix Representation (MR) for graphs of varying sizes and demonstrated its potential properties. Additionally, we explain why MR can preserve graph information.

Once we obtain the feature matrix $Agg(\mathcal{G}(A, X, E))$ of the graph, the function $Pool$ multiplies the transposed feature matrix by the feature matrix itself and provides the graph-level representation as follows,

$$\begin{aligned} Pool(\mathcal{G}) &= Agg(\mathcal{G}(A_{pem}^T, X, E))^T Agg(\mathcal{G}(A_{pem}, X, E)) \\ &= X^T A_{pem}^2 X + X^T A_{pem}E + E^T A_{pem}X + E^T E \end{aligned} \tag{5}$$

Formally, the complete node features and adjacency matrix $A_{pem}$ are retained. Additionally, whether the graph is directed or undirected, $A_{pem}$ and $A_{pem}^2$ have a one-to-one correspondence.

**Lemma 3.2.** *For matrix $A \in \mathbb{R}^{m \times m}$ and $B \in \mathbb{R}^{n \times n}$, if $A$ and $B$ do not have the same eigenvalues, then the solution to the matrix equation $AX = XB$ is $X = 0$.*

According to **Lemma** 3.2, we can prove the following corollary.

**Corollary 3.3.** *For matrix $A \in \mathbb{R}^{n \times n}$ and $B \in \mathbb{R}^{n \times n}$, let the eigenvalues of $A$ be $\lambda_i^A$ for $i = 1, 2, ..., n$, and the eigenvalues of $B$ be $\lambda_i^B$ for $i = 1, 2, ..., n$. If $\forall i \leq n$, $\lambda_i^A > 0$ and $\lambda_i^B > 0$, and $A^2 = B^2$, then $A = B$.*

According to **Proposition** 3.1 and **Corollary** 3.3, $A_{pem}$ and $A_{pem}^2$ have a one-to-one correspondence because all eigenvalues of $A_{pem}$ are positive. Furthermore, $Pool(\mathcal{G})$ is an injective mapping for the input graph if the graph is undirected and the node feature matrix is fixed. In this way, we can learn the connectivity structure of undirect graph without node features.

**Proposition 3.4.** *For undirected graphs with equal and fixed node and edge features, if $\forall i \neq j$, $Pool(\mathcal{G}_i(A_i, X_{fix}, E_{fix})) = Pool(\mathcal{G}_j(A_j, X_{fix}, E_{fix}))$, then $\mathcal{G}_i = \mathcal{G}_j$.*

The equation of $Pool(\mathcal{G})$ 5 shows that $Pool(\mathcal{G})$ performs the aggregation process $Agg(\mathcal{G})$ twice to obtain two feature matrices for directed graphs. Consequently, we modify $Pool(\mathcal{G})$ as follows,

$$Pool(\mathcal{G}) = Agg(\mathcal{G}(A_{pem}, X, E))^T Agg(\mathcal{G}(A_{pem}, X, E)) \tag{6}$$

For undirect graphs, Equation 5 equals Equation 6. For direct graphs, the primary difference lies in converting $A_{pem}^2$ into $A_{pem}^T A_{pem}$. In this way, we have the following proposition.

**Proposition 3.5.** *For matrix $A \in \mathbb{R}^{m \times m}$ and $B \in \mathbb{R}^{n \times n}$ with positive eigenvalues, if $A^T A = B^T B$, then we have $A = QB$, where $Q$ is an orthogonal matrix and $\det(Q) = 1$.*

We then attempt to introduce a neural network to fit orthogonal transformation, thus maintaining the one-to-one correspondence. The aggregation process is calculated as follows:

$$Agg(\mathcal{G}(A_{pem}, X, E)) = \phi^N(A_{pem} X + \phi^E(E)) \tag{7}$$

where $\phi^N : \mathcal{V}^{d_{\mathcal{X}}} \to \mathcal{V}^d$ and $\phi^E : \mathcal{V}^{d_{\varepsilon}} \to \mathcal{V}^{d_{\mathcal{X}}}$ are neural network modules that act on each row of the input matrix. The corresponding $Pool(\mathcal{G})$ is then modified as follows,

$$Pool(\mathcal{G}) = \phi^N(A_{pem} X + \phi^E(E))^T \phi^N(A_{pem} X + \phi^E(E)) \tag{8}$$

Due to the nonlinear changes in neural networks, the theoretical results mentioned above will shift from being deterministic to being existent, meaning the properties depend on the neural network.

The function $Pool(\mathcal{G})$ in Equation 8 has two important properties: permutation invariance and retention of graph information.

**Proposition 3.6.** *If feature matrix of nodes is not fixed, $Pool(\mathcal{G})$ is permutation invariant.*

Moreover, $Pool(\mathcal{G})$ is an effective operation that maintains graph information. To clarify the process, we approach the problem from a geometric perspective. Let $\phi^N(A_{pem} X + \phi^E(E))^T \phi^N(A_{pem} X + \phi^E(E))$ be a linear operator $\psi \in \mathcal{L}(\mathcal{V}^d)$, and $\phi^N(A_{pem} X + \phi^E(E))$ be a linear operator $\eta \in \mathcal{L}(\mathcal{V}^d, \mathcal{U}^n)$. Then, we have $\psi = \eta^T \eta$.

**Proposition 3.7.** *Suppose $\eta \in \mathcal{L}(\mathcal{V}^d, \mathcal{U}^n)$ and $\psi = \eta^T \eta \in \mathcal{L}(\mathcal{V}^d)$. Then, the image space of $\psi$ is isomorphic to that of $\eta$ and there exist a isomorphic mapping $\xi$ that makes $\xi \psi = \eta$.*

Since $Pool(\mathcal{G})$ and $Agg(\mathcal{G})$ map $\mathcal{G}$ into the same geometric space, their representation powers are equal. Then, we perform a linear transformation to provide **the final MR** as follows,

$$Mat(\mathcal{G}) = Agg(\mathcal{G})^T Agg(\mathcal{G}) \odot M \tag{9}$$

where $M = (m_{ij})_{d \times d}$ is a combination of natural base $\in \mathbb{R}^{d \times d}$, conducing as linear transformation.

**Proposition 3.8.** *Let $f(A) = A \odot M$, where $A \in \mathbb{R}^{d \times d}$ and $M \in \mathbb{R}^{d \times d}$. If $\forall i, j \in \{1, 2, ..., d\}$, $M_{i,j} \neq 0$, then $f$ is a isomorphic mapping.*

Therefore, the mapping $Mat : \mathbb{R}^{n_i \times d} \to \mathbb{R}^{d \times d}, \forall n_i \in \mathbb{N}^+$ can provide a unified graph-level representation and preserve the graph information.

$$\begin{bmatrix} W_{11}^L & W_{12}^L & W_{13}^L & W_{14}^L \\ W_{21}^L & W_{22}^L & W_{23}^L & W_{24}^L \\ W_{31}^L & W_{32}^L & W_{33}^L & W_{34}^L \end{bmatrix} \times \begin{bmatrix} v_1^T v_1 m_{11} & v_1^T v_2 m_{12} & v_1^T v_3 m_{13} & v_1^T v_4 m_{14} \\ v_2^T v_1 m_{21} & v_2^T v_2 m_{22} & v_2^T v_3 m_{23} & v_2^T v_4 m_{24} \\ v_3^T v_1 m_{31} & v_3^T v_2 m_{32} & v_3^T v_3 m_{33} & v_3^T v_4 m_{34} \\ v_4^T v_1 m_{41} & v_4^T v_2 m_{42} & v_4^T v_3 m_{43} & v_4^T v_4 m_{44} \end{bmatrix} \times \begin{bmatrix} W_{11}^R & W_{12}^R & W_{13}^R \\ W_{21}^R & W_{22}^R & W_{23}^R \\ W_{31}^R & W_{32}^R & W_{33}^R \\ W_{41}^R & W_{42}^R & W_{43}^R \end{bmatrix}$$

Figure 1: Illustration of Matrix Neural Network.

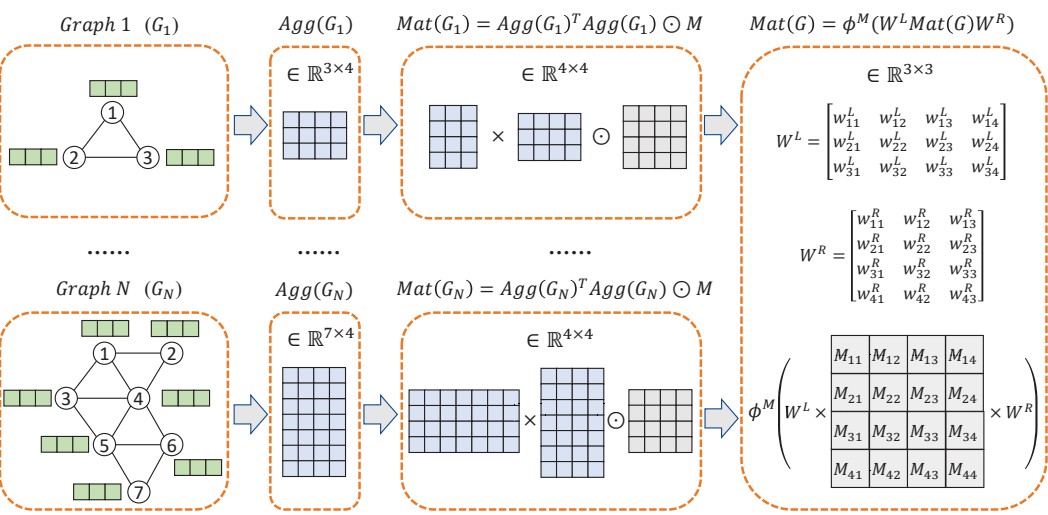

Figure 2: The entire process of MatPool includes the following steps: (i) Utilize PEM to aggregate the node information; (ii) Calculate the graph-level MR; (iii) Feed the graph-level MR into the MNN to extract deeper features for graph property prediction.

## 3.4 MATRIX NEURAL NETWORK

We design a novel Matrix Neural Network (MNN) to extract deeper features from MR, emphasizing row-column correlations caused by multiplication of feature matrix in the function $Mat(\mathcal{G})$.

Assume $Agg(\mathcal{G})$ returns $[v_1, v_2, ..., v_d] \in \mathbb{R}^{n \times d}$ and $d$ is the embedding dimension. Then, $Mat(\mathcal{G})$ can be rewritten in the following matrix form,

$$Mat(\mathcal{G}) = \begin{bmatrix} v_1^T v_1 m_{11} & v_1^T v_2 m_{12} & \dots & v_1^T v_d m_{1d} \\ v_2^T v_1 m_{21} & v_2^T v_2 m_{22} & \dots & v_2^T v_d m_{2d} \\ \vdots & \vdots & \vdots & \vdots \\ v_d^T v_1 m_{d1} & v_d^T v_2 m_{d2} & \dots & v_d^T v_d m_{dd} \end{bmatrix} \tag{10}$$

$Mat(\mathcal{G})$ provides a matrix-based representation where each element is an inner product of paired features. Figure 1 shows the row-column correlation, the element $Mat(\mathcal{G})_{i,j}$ in the matrix are closely related to the elements in the respective $i^{th}$ row and $j^{th}$ column.

To extract deeper features from the MR, we designed a Matrix Neural Network (MNN) that directly processes MR by using two-sided weight matrices. For a matrix representation $Mat(\mathcal{G}) \in \mathbb{R}^{d \times d}$, MNN returns an $output \in \mathbb{R}^{m \times n}$ by performing the following operations,

$$Mat(\mathcal{G}) = \phi^M(W_{m \times d}^L Mat(\mathcal{G})_{d \times d} W_{d \times n}^R) \tag{11}$$

where $\phi^M$ is an activation function applied to each element of the matrix feature. $W_{m \times d}^L \in \mathbb{R}^{m \times d}$ and $W_{d \times n}^R \in \mathbb{R}^{d \times n}$ are two-sided weight matrices acting on the MR. The forward and backward processes of the MNN are detailed in 6 and the MNN has the following properties:

- Since matrix multiplication can run quickly, it provides a significant advantage for MNN in terms of running speed.
- If the output is $\in \mathbb{R}^{d \times d}$, the MNN only requires $2d^2$ parameters to handle a matrix-pattern-oriented feature $\in \mathbb{R}^{d \times d}$.
- The MNN naturally constrains the rows and columns of $Mat(\mathcal{G})$ through the columns of $W^R$ and the rows of $W^L$, respectively.

The framework of the MatPool is shown in Figure 2, and its pseudo-code is listed in Algorithm 1. According to the pseudo-code, suppose the number of GNN layers, nodes, edges, and feature dimensions in one graph are $L$, $n$, $e$, and $d$ respectively. The time complexities of PEM, MR, and MNN are $O(Ln^2d + Lned + Lnd^2)$, $O(nd^2)$, and $O(d^3)$, respectively. Therefore, the primary time complexity is concentrated on the message-passing processes.

## 4 EXPERIMENT

In this section, we validate the effectiveness and efficiency of MatPool through extensive experiments. The computations are performed on a computer with an Intel i9 12900K processor and an RTX A6000 GPU. In the experiment, we will address the following questions:

- **The Effectiveness of Positive Eigenvalue Mapping (PEM)**: Why is PEM used as the propagating and aggregating way to calculate the feature matrix for providing the Matrix Representation (MR)?
- **The Effectiveness of Matrix Neural Network (MNN)**: Why MNN is used as the neural network structure to extract deeper features from the MR?
- **The Effectiveness of MatPool**: How does MatPool compare to other pooling methods for graph property prediction?
- **The efficiency of MatPool**: Does MatPool offer advantages in training speed for graph property prediction?

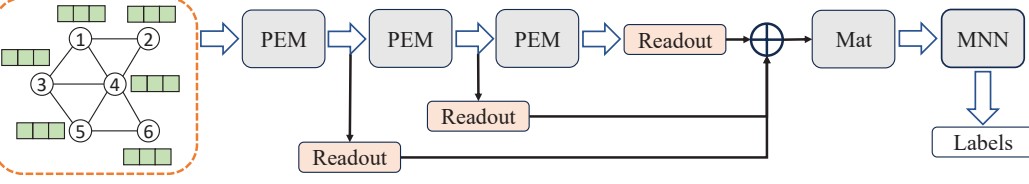

Figure 3: The algorithm flowchart of MatPool

Figure 3 shows the flowchart corresponding to the pseudo-code of MatPool. In the implementation, we accumulate the feature matrix mapped may PEM layer by layer. Then, we conduct $Mat((G))$ that multiplies the transpose of accumulated feature matrix by the feature matrix itself and a Hadamard product to provide a unified MR for each graph. Next, MNN extracts the deeper feature from MR, followed by a neural network to predict the final label.

### 4.1 EXPERIMENT SETTING

**Used Datasets**: We compare the experimental results of all algorithms on 20 widely used graph datasets. Four of these datasets are from the OGBG graph datasets Hu et al. (2021), which naturally divide the training, validation, and testing sets. Since most values in the node features of PPA are zero, edge features are necessary for conducting experiments on PPA. The remaining 16 datasets are

Table 2: Comparison results (%) of the combination of propagating and aggregating way and neural network structure. (The best result on each data set is written in bold).

| Name | MOLHIV | | MOLPCBA | | PPA | CODE2 | | Average |
|------|--------|------|---------|------|-----|-------|------|---------|
| Edge Feature | $w/$ | $w/o$ | $w/$ | $w/o$ | $w/$ | $w/$ | $w/o$ | |
| **PEM+MNN** | $\mathbf{79.2}_{\pm0.7}$ | $\mathbf{79.5}_{\pm1.1}$ | $24.0_{\pm0.5}$ | $24.1_{\pm0.3}$ | $\mathbf{71.3}_{\pm0.5}$ | $15.5_{\pm0.9}$ | $\mathbf{16.0}_{\pm0.6}$ | $\mathbf{44.2}_{\pm0.7}$ |
| GCN+**MNN** | $78.0_{\pm1.9}$ | $76.9_{\pm1.3}$ | $23.4_{\pm0.2}$ | $23.3_{\pm0.4}$ | $63.5_{\pm4.9}$ | $\mathbf{16.0}_{\pm0.6}$ | $16.0_{\pm0.3}$ | $42.4_{\pm1.4}$ |
| GIN+**MNN** | $78.4_{\pm1.3}$ | $78.9_{\pm1.4}$ | $\mathbf{24.5}_{\pm0.3}$ | $\mathbf{24.6}_{\pm0.3}$ | $66.5_{\pm1.3}$ | $15.6_{\pm1.0}$ | $15.5_{\pm1.1}$ | $43.4_{\pm0.9}$ |
| **PEM+MLP** | $77.4_{\pm1.6}$ | $77.8_{\pm0.9}$ | $23.0_{\pm0.3}$ | $23.0_{\pm0.4}$ | $71.1_{\pm0.7}$ | $14.2_{\pm0.3}$ | $14.4_{\pm0.4}$ | $43.0_{\pm0.6}$ |
| **PEM+CNN** | $79.1_{\pm1.0}$ | $79.3_{\pm1.3}$ | $22.8_{\pm0.5}$ | $22.8_{\pm1.3}$ | $52.2_{\pm19.4}$ | $13.1_{\pm0.2}$ | $13.0_{\pm0.3}$ | $40.3_{\pm3.4}$ |

from TUDataset (Morris et al., 2020). The performance of the methods is tested using 10-fold cross-validation, with one fold for validation, one for testing, and the remaining for training. Detailed descriptions of all datasets are provided in 5. Because some TUDataset datasets lack attributes, we add degree as a feature for all datasets in TUDataset.

**Basic Setting**: In the experiment, we validate the effectiveness of PEM and then adopt PEM as the backbone to test the effectiveness of all pooling methods. The descriptions of all hyper-parameters are listed in 6. We conduct experiments on each combination of hyper-parameters 10 times, averaging the results to obtain the final outcome. For the TUDataset, we select the best learning rate on the validation set to predict the test set. Adam Kingma & Ba (2015) is selected as the optimizer. The learning rate begins to decay after 20 epochs at a rate of 0.95. We stop the training process early if there is no improvement for 15 epochs on OGBG datasets and 20 epochs on TUDatasets.

**Comparison Methods**: We have selected eight pooling methods categorized into global and hierarchical pooling as comparison methods. The global pooling methods include Global Attention (GA) Li et al. (2016), Set2Set Vinyals et al. (2016), Memory-based Pooling (MEN) Khasahmadi et al. (2020), and Second-Order Pooling (SOPool) Wang & Ji (2023). Hierarchical pooling methods include TopK Pooling Gao & Ji (2019), Self-Attention Pooling (SAG) Lee et al. (2019), Path Integral Based Pooling (PAN) Ma et al. (2020) and Adaptive Structure Aware Pooling (ASAP) Ranjan et al. (2020). The eight pooling methods can be easily callable into the framework of MatPool.

Moreover, we compare MatPool with other important baselines such as DiffPool Ying et al. (2018), MuchPool , GMT Baek et al. (2021), StructPool Yuan & Ji (2020b), MinCutPool Bianchi et al. (2020a), DKEPool Chen et al. (2023), and SortPool Zhang et al. (2018). The settings of datasets and algorithms follow that in GMT, and the results of the comparison algorithms are directly copied from GMT. The experimental results can be seen in Table 7 in the Appendix.

### 4.2 PERFORMANCE OF POSITIVE EIGENVALUE MAPPING AND MATRIX NEURAL NETWORK

This experiment on large-scale graph datasets demonstrates that PEM and MNN are essential components of MatPool. Replacing either module results in a decline in performance.

Table 2 shows that using both PEM and MNN achieves the best performance on most OGBG datasets. When the MNN module is fixed, PEM achieves the best performance on 4 out of 7 datasets and the highest average performance, demonstrating its superiority for graph property prediction.

When we fix the PEM module and use MLP, we compress the large matrix and flatten it into a vector for graph property prediction. Additionally, we use AlexNet (Krizhevsky et al., 2012) as the CNN model. The scale of parameters in MLP and CNN are similar to that in MNN. From the table, it is clear that MNN has a significant performance advantage over the other datasets, demonstrating that MNN surpasses both MLP and CNN comprehensively.

When the message-passing model is combined with MNN, PEM focuses on key nodes and achieves the best results. GIN treats all nodes equally, doing slightly better than PEM on MOLPCBA but worse on other datasets. GCN, which adjusts the adjacency matrix based on node degrees, performs the worst overall.

From Table 3, it is evident that the Hadamard product operation $\odot M$ plays an important role in MatPool. Regardless of whether the subsequent networks are MNN, CNN, or MLP, the operation

Table 3: Comparison results (%) of MatPool with and without the operation of $\odot M$. (The improved result on each data set is written in bold).

| Name | MOLHIV | | MOLPCBA | | PPA | CODE2 | | Average |
|---|---|---|---|---|---|---|---|---|
| Edge Feature | w/ | w/o | w/ | w/o | w/ | w/ | w/o | |
| PEM+MNN | $78.4_{\pm1.3}$ | $78.5_{\pm1.4}$ | $21.9_{\pm0.5}$ | $21.9_{\pm0.7}$ | $66.4_{\pm1.2}$ | $14.9_{\pm0.6}$ | $15.4_{\pm0.6}$ | $42.5_{\pm0.9}$ |
| PEM+$\odot$M+MNN | $79.2_{\pm0.7}$ | $79.5_{\pm1.1}$ | $24.0_{\pm0.5}$ | $24.1_{\pm0.3}$ | $71.3_{\pm0.5}$ | $15.5_{\pm0.9}$ | $16.0_{\pm0.6}$ | $44.2_{\pm0.7}$ |
| $\odot$M Improvement | **0.8** ↑ | **1** ↑ | **2.1** ↑ | **2.2** ↑ | **4.9** ↑ | **0.6** ↑ | **0.6** ↑ | **1.7** ↑ |
| PEM+CNN | $78.3_{\pm1.5}$ | $78.9_{\pm1.4}$ | $22.1_{\pm4.2}$ | $21.5_{\pm4.1}$ | $47.3_{\pm20.8}$ | $12.9_{\pm0.3}$ | $13.0_{\pm0.2}$ | $39.1_{\pm4.6}$ |
| PEM+$\odot$M+CNN | $79.1_{\pm1.0}$ | $79.3_{\pm1.3}$ | $22.8_{\pm0.5}$ | $22.8_{\pm1.3}$ | $52.2_{\pm19.4}$ | $13.1_{\pm0.2}$ | $13.0_{\pm0.3}$ | $40.3_{\pm3.4}$ |
| $\odot$M Improvement | **0.8** ↑ | **0.4** ↑ | **0.7** ↑ | **1.3** ↑ | **4.9** ↑ | **0.2** ↑ | 0 | **1.2** ↑ |
| PEM+MLP | $77.3_{\pm1.2}$ | $77.2_{\pm1.4}$ | $22.8_{\pm0.4}$ | $22.8_{\pm0.3}$ | $67.8_{\pm0.7}$ | $14.6_{\pm0.2}$ | $14.6_{\pm0.2}$ | $42.4_{\pm0.6}$ |
| PEM+$\odot$M+MLP | $77.4_{\pm1.6}$ | $77.8_{\pm0.9}$ | $23.0_{\pm0.3}$ | $23.0_{\pm0.4}$ | $71.1_{\pm0.7}$ | $14.2_{\pm0.3}$ | $14.4_{\pm0.4}$ | $43.0_{\pm0.6}$ |
| $\odot$M Improvement | **0.1** ↑ | **0.6** ↑ | **0.2** ↑ | **0.2** ↑ | **3.3** ↑ | -0.4 | -0.2 | **0.6** ↑ |

$\odot M$ improves the results. This improvement is more significant with MNN because MNN operates on the entire matrix. CNN operates on local matrices, while MLP destroys the matrix structure after flattening. Therefore, the improvement on CNN and MLP is slightly smaller than that on MNN.

In summary, for feature representations with row-column correlations, MNN can extract deeper features and achieve better experimental results. MLP generally converts the matrix form into a vector form, destroying the matrix structure and resulting in a high-dimensional vector. Although CNN can process matrix features like images, it fails to capture the row-column correlation effectively.

## 4.3 PERFORMANCE OF MATPOOL FOR GRAPH PROPERTY PREDICTION

This experiment on graph datasets demonstrates that MatPool outperforms other easily callable pooling methods for graph property prediction.

Table 4: Experimental results (%) for all pooling methods using PEM as the message-passing way are reported here (The best result on each data set is written in bold).

| Name | MatPool | SOPool | GA | Set2Set | MEM | TopK | SAG | PAN | ASAP |
|---|---|---|---|---|---|---|---|---|---|
| MOLHIV | $\mathbf{79.5}_{\pm1.1}$ | $78.7_{\pm0.9}$ | $75.9_{\pm2.4}$ | $74.9_{\pm2.2}$ | $78.6_{\pm1.2}$ | $74.0_{\pm1.9}$ | $74.0_{\pm3.2}$ | $73.5_{\pm2.2}$ | $73.5_{\pm2.1}$ |
| MOLPCBA | $\mathbf{24.1}_{\pm0.3}$ | $20.5_{\pm1.8}$ | $22.3_{\pm0.4}$ | $21.5_{\pm0.6}$ | $23.8_{\pm0.3}$ | $17.2_{\pm2.5}$ | $18.5_{\pm0.8}$ | $15.1_{\pm0.3}$ | $19.6_{\pm1.2}$ |
| PPA | $71.3_{\pm0.5}$ | $33.2_{\pm19}$ | $33.0_{\pm8.4}$ | $\mathbf{71.8}_{\pm2.1}$ | $64.6_{\pm16}$ | $54.3_{\pm21}$ | $67.0_{\pm1.9}$ | $69.7_{\pm0.8}$ | *OOT* |
| CODE2 | $\mathbf{16.0}_{\pm0.7}$ | $12.7_{\pm3.1}$ | $15.5_{\pm0.7}$ | $15.3_{\pm0.4}$ | $14.0_{\pm0.4}$ | $14.5_{\pm0.4}$ | $15.0_{\pm0.6}$ | $14.4_{\pm0.7}$ | *OOT* |
| AIDS | $99.0_{\pm0.1}$ | $99.2_{\pm0.4}$ | $98.6_{\pm0.1}$ | $98.7_{\pm0.2}$ | $\mathbf{99.5}_{\pm0.2}$ | $99.0_{\pm0.2}$ | $99.0_{\pm0.3}$ | $98.6_{\pm0.2}$ | $98.8_{\pm0.2}$ |
| FRANKENSTEIN | $73.9_{\pm0.5}$ | $72.2_{\pm0.8}$ | $73.8_{\pm0.5}$ | $72.6_{\pm0.7}$ | $\mathbf{74.1}_{\pm0.5}$ | $71.6_{\pm0.9}$ | $71.7_{\pm0.9}$ | $69.8_{\pm0.8}$ | $70.2_{\pm0.9}$ |
| MUTAGENICITY | $82.4_{\pm0.4}$ | $82.3_{\pm0.4}$ | $\mathbf{82.8}_{\pm0.3}$ | $81.9_{\pm0.4}$ | $82.5_{\pm0.4}$ | $78.7_{\pm1.1}$ | $79.0_{\pm1.1}$ | $80.6_{\pm0.5}$ | $78.2_{\pm1.2}$ |
| NCI1 | $\mathbf{81.5}_{\pm0.4}$ | $80.7_{\pm0.4}$ | $81.1_{\pm0.4}$ | $80.3_{\pm0.5}$ | $81.4_{\pm0.7}$ | $77.3_{\pm0.9}$ | $77.8_{\pm0.6}$ | $76.9_{\pm0.6}$ | $77.8_{\pm1.0}$ |
| NCI109 | $\mathbf{80.5}_{\pm0.4}$ | $79.2_{\pm0.5}$ | $79.8_{\pm0.5}$ | $79.7_{\pm0.6}$ | $80.3_{\pm0.4}$ | $76.4_{\pm1.1}$ | $77.0_{\pm0.9}$ | $76.1_{\pm0.7}$ | $76.7_{\pm1.2}$ |
| DD | $75.6_{\pm0.7}$ | $76.0_{\pm0.4}$ | $67.6_{\pm1.1}$ | $71.1_{\pm1.2}$ | $\mathbf{76.4}_{\pm0.4}$ | $74.7_{\pm1.1}$ | $74.4_{\pm1.0}$ | $74.6_{\pm0.8}$ | $74.1_{\pm0.9}$ |
| PROTEINS | $\mathbf{75.1}_{\pm0.8}$ | $74.9_{\pm0.8}$ | $71.8_{\pm0.8}$ | $70.8_{\pm1.2}$ | $74.6_{\pm0.5}$ | $73.8_{\pm0.5}$ | $73.6_{\pm0.5}$ | $74.6_{\pm0.8}$ | $74.2_{\pm1.1}$ |
| COIL-DEL | $\mathbf{83.9}_{\pm0.3}$ | $76.6_{\pm0.6}$ | $81.7_{\pm0.6}$ | $81.5_{\pm0.6}$ | $79.1_{\pm0.4}$ | $71.7_{\pm0.6}$ | $69.5_{\pm0.5}$ | $70.0_{\pm0.7}$ | $75.5_{\pm0.8}$ |
| COIL-RAG | $95.9_{\pm0.4}$ | $95.4_{\pm0.3}$ | $95.8_{\pm0.3}$ | $\mathbf{97.0}_{\pm0.2}$ | $96.0_{\pm0.3}$ | $94.9_{\pm0.4}$ | $95.1_{\pm0.2}$ | $95.4_{\pm0.2}$ | $95.8_{\pm0.3}$ |
| Letter-high | $89.5_{\pm0.5}$ | $87.7_{\pm0.5}$ | $87.7_{\pm0.6}$ | $89.4_{\pm0.6}$ | $89.2_{\pm0.4}$ | $82.8_{\pm0.8}$ | $85.4_{\pm1.0}$ | $\mathbf{93.3}_{\pm0.4}$ | $86.9_{\pm0.8}$ |
| Letter-low | $\mathbf{98.4}_{\pm0.2}$ | $97.8_{\pm0.4}$ | $98.1_{\pm0.2}$ | $98.0_{\pm0.2}$ | $98.2_{\pm0.3}$ | $96.7_{\pm0.4}$ | $96.7_{\pm0.3}$ | $98.0_{\pm0.2}$ | $97.1_{\pm1.2}$ |
| Letter-med | $93.4_{\pm0.4}$ | $92.1_{\pm0.5}$ | $92.0_{\pm0.6}$ | $92.5_{\pm0.3}$ | $93.3_{\pm0.5}$ | $88.9_{\pm0.3}$ | $89.7_{\pm0.7}$ | $\mathbf{95.6}_{\pm0.4}$ | $90.7_{\pm0.4}$ |
| COLLAB | $\mathbf{81.7}_{\pm0.5}$ | $80.3_{\pm0.8}$ | $79.1_{\pm1.4}$ | $80.9_{\pm0.9}$ | $78.6_{\pm3.6}$ | $77.9_{\pm1.2}$ | $77.7_{\pm1.3}$ | $79.0_{\pm0.8}$ | $65.0_{\pm3.4}$ |
| IMDB-BINARY | $72.9_{\pm1.2}$ | $72.2_{\pm1.3}$ | $72.6_{\pm1.0}$ | $70.3_{\pm1.3}$ | $\mathbf{73.6}_{\pm1.0}$ | $71.2_{\pm1.0}$ | $71.2_{\pm0.9}$ | $70.3_{\pm0.7}$ | $70.9_{\pm1.3}$ |
| IMDB-MULTI | $\mathbf{50.3}_{\pm0.7}$ | $49.8_{\pm0.6}$ | $48.8_{\pm0.9}$ | $46.8_{\pm1.9}$ | $50.0_{\pm0.8}$ | $48.6_{\pm0.7}$ | $48.9_{\pm0.9}$ | $49.0_{\pm1.2}$ | $48.7_{\pm0.8}$ |
| COLORS-3 | $\mathbf{100.0}_{\pm0.0}$ | $99.2_{\pm0.9}$ | $36.1_{\pm1.4}$ | $51.1_{\pm1.1}$ | $\mathbf{100.0}_{\pm0.0}$ | $65.7_{\pm3.0}$ | $61.2_{\pm1.9}$ | $52.8_{\pm1.8}$ | $58.1_{\pm2.8}$ |
| Average | $\mathbf{76.2}_{\pm0.5}$ | $73.0_{\pm1.7}$ | $69.7_{\pm1.1}$ | $72.3_{\pm0.9}$ | $75.4_{\pm1.5}$ | $70.5_{\pm2.0}$ | $71.1_{\pm1.0}$ | $71.3_{\pm0.7}$ | *NA* |

Table 4 shows the experimental results of all pooling methods across the graph datasets used. MatPool achieves the highest results on 11 out of 20 datasets and has the highest average result. Overall, global pooling methods outperform hierarchical pooling methods, and MatPool performs better than other global pooling methods.

The Friedman test Iman & Davenport (1980); Nemenyi (1963) and Bayesian signed-rank test analysis Benavoli et al. (2016) shown in 4 indicate that MatPool achieves the highest ranking. Except for MEM, other methods are incomparable to MatPool. Additionally, the heatmap of MatPool demonstrates an absolute advantage over other graph pooling methods with a probability of nearly 100%.

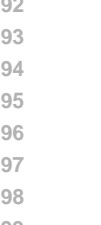 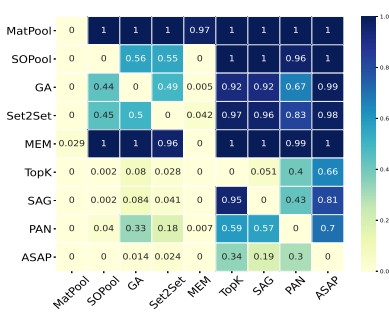

(a) Friedmen test result      (b) Heat-map of Bayesian signed-rank test

Figure 4: Statical results by using Friedman test and Bayesian signed-rank test.

### 4.4 TRAINING EFFICIENCY FOR GRAPH PROPERTY PREDICTION

This experiment demonstrates that MatPool, as a global pooling method, offers the advantages of simple implementation and fast training speed.

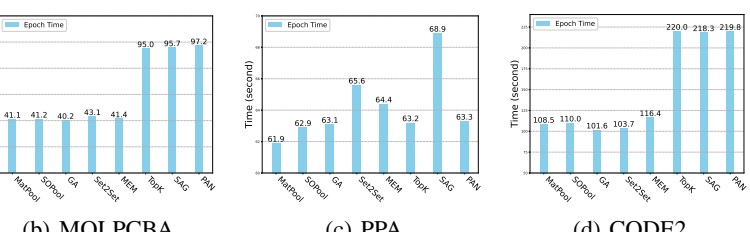

(a) MOLHIV      (b) MOLPCBA      (c) PPA      (d) CODE2

Figure 5: Training time of all pooling methods on OGBG datasets.

In our experiment, all global pooling methods have a similar scale of parameters. Following the example of MNN, the other pooling methods also use two heads to map the features. From Figure 5, all global pooling methods have similar training times, while hierarchical pooling methods require significantly more time due to the need for additional operations in the layer-by-layer pooling.

### 5 CONCLUSION

Unlike existing graph pooling methods that provide graph-level representations based on vector patterns and loss graph information, we explore the use of matrix patterns and propose a new method named MatPool for representing and predicting graphs of different size. MatPool consists of three main components: Positive Eigenvalue Mapping (PEM), Matrix Representation (MR), and Matrix Neural Network (MNN). PEM reconstructs the adjacency matrix to have positive eigenvalues, enhancing the propagation ability of primary nodes. MR provides a unified matrix-pattern-oriented representation with key properties such as permutation invariance and retention of graph information. MNN is specifically designed to extract deeper features from the row-column correlated MR. We have theoretically analyzed the properties of MatPool and conducted extensive experiments to validate its efficiency and effectiveness in graph properties prediction.

The main drawback of MatPool lies in the process of multiplying the feature matrix by its transpose. Each feature in MR is obtained by summing the squares, which may result in significant numerical variation. Therefore, a more reasonable normalization process and careful initialization of values in the MNN network are needed in the future.

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

# A  APPENDIX

This appendix provides theoretical and experimental materials and is organized as follows: Subsection A.1 contains detailed proofs, including six propositions, one corollary, and one lemma of MatPool. It also outlines the forward and backward processes of MNN. Subsection A.2 provides the pseudo-code and detailed time complexity analysis of MatPool. Subsection A.3 offers detailed dataset descriptions and hyper-parameter settings. Subsection A.4 presents additional results, including the convergence of MatPool and the effectiveness of hyper-parameters. Subsection A.5 presents comparison results between MatPool and other classical pooling methods.

## A.1  THEORETICAL PROOFS OF MATPOOL AND GRADIENT CALCULATION OF MNN

**Proposition 3.1.** *Given an adjacency matrix $A \in \mathbb{R}^{n \times n}$, let $\lambda_i$ be the eigenvalue of the reconstructed adjacency matrix $A_{pem}$. Then, $\forall i \leq n, \lambda_i > 0$.*

*Proof.* Each diagonal element in the reconstructed adjacency matrix $A_{pem}$ is positive. $\forall i = 1, 2, ..., n$, we have the following equation,

$$a_{ii} > \sum_{j=1, j \neq i}^{n} |a_{ij}| = \sum_{j=1, j \neq i}^{n} a_{ij}. \tag{12}$$

Therefore, $A_{pdm}$ is a diagonally dominant matrix. Based on the Gerschgorin Circle Theorem and the property of the eigenvalue of the matrix, we have the following equation, indicating that the eigenvalues of $A_{pem}$ fall within the following circle,

$$|z - a_{ii}| \leq R_i = \sum_{j=1, j \neq i}^{n} a_{ij} \tag{13}$$

Consequently, $\forall i \leq n, \lambda_i > 0$, and we can conclude that each eigenvalue of $A_{pem}$ is positive. □

**Lemma 3.2.** *For matrix $A \in \mathbb{R}^{m \times m}$ and $B \in \mathbb{R}^{n \times n}$, if $A$ and $B$ do not have the same eigenvalues, then the solution to the matrix equation $AX = XB$ is $X = 0$.*

*Proof.* Assume that $f(\lambda) = |\lambda I - A|$ is the characteristic polynomial of $A$. According to Cayley-Hanmilton Theorem, we have,

$$f(A) = 0 \tag{14}$$

Next, we modify $AX = XB$ as follows,

$$A^2 X = A(AX) = A(XB) = (AX)B = XB^2 \tag{15}$$

Then, we can get the following equation,

$$f(A)X = Xf(B) = 0 \tag{16}$$

Assume that the eigenvalues of $B$ are $\mu_1, \mu_2, ..., \mu_n$, and the eigenvalues of $f(B)$ are $f(\mu_1), f(\mu_2), ..., f(\mu_n)$.

Assert: $\mu_1, \mu_2, ..., \mu_n$ are not the eigenvalues of $A$.

If $\exists \mu_i$ such that $f(\mu_i) = 0$, then $\mu_i$ is an eigenvalue of $A$. However, $A$ and $B$ do not share the same eigenvalues. Therefore, $\forall i, \mu_i$ is not a root of the characteristic polynomial $f(A)$, and we can conclude that $f(B)$ is an invertible matrix.

Finally, we can calculate the solution of the matrix equation $AX = XB$ as follows,

$$Xf(B) = 0 \tag{17}$$

$$X = 0f(B)^{-1} = 0 \tag{18}$$

□

**Corollary 3.3.** *For matrix $A \in \mathbb{R}^{n \times n}$ and $B \in \mathbb{R}^{n \times n}$, let the eigenvalues of $A$ be $\lambda_i^A$ for $i = 1, 2, ..., n$, and the eigenvalues of $B$ be $\lambda_i^B$ for $i = 1, 2, ..., n$. If $\forall i \leq n$, $\lambda_i^A > 0$ and $\lambda_i^B > 0$, and $A^2 = B^2$, then $A = B$.*

*Proof.* Proving $A = B$ is equivalent to proving $A - B = 0$. Therefore, we present the following equation,

$$A(A - B) = A^2 - AB = B^2 - AB = (A - B)(-B) \tag{19}$$

Because $\forall i \leq n$, $\lambda_i^A > 0$ and $\lambda_i^B > 0$, the eigenvalues of $A$ are positive while those of $-B$ are negative, meaning that $A$ and $-B$ do not share the same eigenvalues. According to **Lemma** 3.2, we can conclude that,

$$A - B = 0 \tag{20}$$

Therefore, we conclude that $A_{pem}$ and $A_{pem}^2$ have one-to-one correspondence for undirect graphs. $\square$

**Proposition 3.4.** *For undirected graphs with equal and fixed node and edge features, if $\forall i \neq j, Pool(\mathcal{G}_i(A_i, X_{fix}, E_{fix})) = Pool(\mathcal{G}_j(A_j, X_{fix}, E_{fix}))$, then $\mathcal{G}_i = \mathcal{G}_j$.*

*Proof.* Assume the node feature matrix $X_{fix}$ is fixed for all graphs, and adjacency matrices of $\mathcal{G}_i$ and $\mathcal{G}_j$ are $A_i = A$ and $A_j = B$, respectively. Then, proving $Pool(\mathcal{G}_i(A_i, X_{fix}, E_{fix})) = Pool(\mathcal{G}_j(A_j, X_{fix}, E_{fix}))$ is equivalent to proving follows,

$$Pool(\mathcal{G}_i) = X_{fix}^T A X_{fix} = X_{fix}^T B X_{fix} = Pool(\mathcal{G}_j) \tag{21}$$

Let $\alpha = \alpha_i + \alpha_j$, where $\alpha_i$ and $\alpha_j$ are unit column vectors. Since $A = (a_{ij})$ is a real symmetric matrix, if $Pool(\mathcal{G}_i) = 0$, we obtain the following equation,

$$a_{ii} = \alpha_i^T A \alpha_i = 0, \tag{22}$$

Moreover, we have the following equations,

$$a_{ij} + a_{ji} = \alpha_i^T A \alpha_j + \alpha_j^T A \alpha_i = \alpha_i^T A \alpha_i + \alpha_j^T A \alpha_j + \alpha_i^T A \alpha_j + \alpha_j^T A \alpha_i = (\alpha_i + \alpha_j)^T A (\alpha_i + \alpha_j) = 0 \tag{23}$$

Since $a_{ij} = a_{ji}$, we can conclude that $a_{ij} = 0$. This leads to $A = 0$, meaning that the kernel space of $Pool$ function $Ker(Pool) = 0$. Therefore, $Pool(\mathcal{G}(A, X_{fix}, E_{fix}))$ and the adjacency matrix $A$ are in one-to-one correspondence when the node features are fixed for all graphs.

In the implementation, the multiplication operation ensures the size of the adjacency matrix is consistent by padding zeros to the right and bottom, which does not cause substantial changes to the matrix. $\square$

**Proposition 3.5.** *For matrix $A \in \mathbb{R}^{m \times m}$ and $B \in \mathbb{R}^{n \times n}$ with positive eigenvalues, if $A^T A = B^T B$, then we have $A = QB$, where $Q$ is an orthogonal matrix and $det(Q) = 1$..*

*Proof.* In MatPool, the eigenvalues of $A_{pem}$ are positive. Therefore, we have $det(A) > 0$ and $det(B) > 0$. This means $A$ and $B$ are an invertible matrices, and we have,

$$A = (A^T)^{-1} B^T B = QB \tag{24}$$

Next, we have

$$\begin{aligned} QQ^T &= (A^T)^{-1} B^T ((A^T)^{-1} B^T)^T \\ &= (A^T)^{-1} B^T B ((A^T)^{-1})^T \\ &= (A^T)^{-1} A^T A A^{-1} \\ &= I \end{aligned} \tag{25}$$

Therefore, $Q$ is an orthogonal matrix. According to the following equation,

$$det(A) = det(QB) = det(Q)det(B) \tag{26}$$

Because $det(A) > 0$ and $det(B) > 0$, we have $det(Q) = 1$, eliminating the possibility of mirror transformation.. $\square$

**Proposition 3.6.** *If feature matrix of nodes is not fixed, $Pool(\mathcal{G})$ is permutation invariant.*

*Proof.* According to the Equation 8, we have,

$$Pool(\mathcal{G}) = Agg(\mathcal{G})^T Agg(\mathcal{G}) = \phi^N(A_{pem}X + \phi^E(E))^T \phi^N(A_{pem}X + \phi^E(E)) \quad (27)$$

where $\phi^N : \mathcal{V}^{d_x} \to \mathcal{V}^d$ and $\phi^E : \mathcal{V}^{d_\varepsilon} \to \mathcal{V}^{d_x}$ are neural network modules that act on each row of the input matrix.

Here, we define the permutation operation $P_{ij}$ as swapping the $i^{th}$ row and the $j^{th}$ row of the matrix, meaning swap two nodes in the input graph and $P_{ij}^T P_{ij} = I$. If we randomly swap the $i^{th}$ row and the $j^{th}$ row of the graph $\mathcal{G}$, the adjacency matrix and the feature matrix will be modified accordingly. Then, the process of $Pool(\mathcal{G})$ can be calculated as follows,

$$\begin{aligned}
Pool(\mathcal{G}) &= \phi^N(P_{ij}AP_{ij}^T P_{ij}X + \phi^E(P_{ij}E))^T \phi^N(P_{ij}AP_{ij}^T P_{ij}X + \phi^E(P_{ij}E)) \\
&= \phi^N(P_{ij}AX + P_{ij}\phi^E(E))^T \phi^N(P_{ij}AX + P_{ij}\phi^E(E))
\end{aligned} \quad (28)$$

Since the neural network mapping functions $\phi^N$ and $\phi^E$ operate on individual rows, the swapping operation $P_{ij}$ does not affect the results and can be factored out of the mapping functions. The equation for $Pool(\mathcal{G})$ can be calculated as follows,

$$\begin{aligned}
Pool(\mathcal{G}) &= \phi^N(AX + \phi^E(E))^T P_{ij}^T P_{ij} \phi^N(AX + \phi^E(E)) \\
&= \phi^N(AX + \phi^E(E))^T \phi^N(AX + \phi^E(E))
\end{aligned} \quad (29)$$

Therefore, all permutation operations are cancelled out in this function, meaning that that $Pool(\mathcal{G})$ is permutation invariant. $Mat(\mathcal{G})$ also inherits the permutation invariance. $\square$

**Proposition 3.7.** *Suppose $\eta \in \mathcal{L}(\mathcal{V}^d, \mathcal{U}^n)$ and $\psi = \eta^T \eta \in \mathcal{L}(\mathcal{V}^d)$. Then, the image space of $\psi$ is isomorphic to that of $\eta$ and there exist a isomorphic mapping $\xi$ that makes $\xi\psi = \eta$.*

*Proof.* Suppose a vector $x \in \mathbb{R}^d$, if the linear operator $\eta$ acts on $x$ , we have,

$$\eta(x) = 0 \Rightarrow \eta^T \eta(x) = 0 \quad (30)$$

Moreover, we have,

$$\begin{aligned}
\eta^T \eta(x) = 0 &\Rightarrow (\eta(x))^T \eta(x) = 0 \\
&\Rightarrow ||\eta(x)|| = 0 \\
&\Rightarrow \eta(x) = 0
\end{aligned} \quad (31)$$

Therefore, we conclude that the kernel space of $\eta$ equals that of $\psi = \eta^T \eta$, meaning $Ker(\psi) = Ker(\eta)$. Thus, we have

$$dimIm(\psi) = d - dimKer(\psi) = d - dimKer(\eta) = dimIm(\eta) \quad (32)$$

Therefore, there exist an isomorphic mapping $\xi : Im(\psi) \to Im(\eta)$, such that $\xi\psi = \xi\eta^T \eta = \eta$. $\square$

**Proposition 3.8.** *Let $f(A) = A \odot M$, where $A \in \mathbb{R}^{d \times d}$ and $M \in \mathbb{R}^{d \times d}$. If $\forall i, j \in \{1, 2, ..., d\}$, $M_{i,j} \neq 0$, then $f$ is a isomorphic mapping.*

*Proof.* $\forall A \in \mathbb{R}^{d \times d}, B \in \mathbb{R}^{d \times d}$, we have

$$f(A + B) = (A + B) \odot M = A \odot M + B \odot M = f(A) + f(B) \quad (33)$$

Moreover, $\forall k \in \mathbb{R}$, we have

$$f(kA) = (kA) \odot M = k(A \odot M) = kf(A) \quad (34)$$

Therefore, $f$ is a linear transformation. Moreover, if $\forall i, j \in \{1, 2, ..., d\}$, $M_{i,j} \neq 0$, we can the relationship between $A_{i,j}$ and $f(A)_{ij}$ is one-to-one. In this way, the function $f(A) = A \odot M, for all i, j \in 1, 2, ..., d, M_{i,j} \neq 0$ is a linear isomorphism. $\square$

Figure 6: Illustration of the forward process of MNN.

**Forward and backward propgation of Matrix Neural Network (MNN).** Figure 6 presents an example that we conduct MNN on a matrix representation $Mat(\mathcal{G}) \in \mathbb{R}^{2 \times 2}$.

Taking Figure 6 as an example of the calculation process of MNN. The forward propagation that calculates the output $M'_{ij}, \forall i, j$ is as follows,

$$M'_{11} = \sum_{k=1}^{2} \sum_{l=1}^{2} W_{1k}^{L} M_{kl} W_{l1}^{R} \tag{35}$$

$$M'_{12} = \sum_{k=1}^{2} \sum_{l=1}^{2} W_{1k}^{L} M_{kl} W_{l2}^{R} \tag{36}$$

$$M'_{21} = \sum_{k=1}^{2} \sum_{l=1}^{2} W_{2k}^{L} M_{kl} W_{l1}^{R} \tag{37}$$

$$M'_{22} = \sum_{k=1}^{2} \sum_{l=1}^{2} W_{2k}^{L} M_{kl} W_{l2}^{R} \tag{38}$$

Suppose the loss value for the graph property prediction task is $L$. According to the chain rule, the partial derivative of $L$ with respect to $W_{ij}^{L}$ can be calculated as follows,

$$\frac{\partial L}{\partial W_{11}^{L}} = \frac{\partial L}{\partial M'_{11}} \frac{\partial M'_{11}}{\partial W_{11}^{L}} + \frac{\partial L}{\partial M'_{12}} \frac{\partial M'_{12}}{\partial W_{11}^{L}} = \frac{\partial L}{\partial M'_{11}} (\sum_{l=1}^{2} M_{1l} W_{l1}^{R}) + \frac{\partial L}{\partial M'_{12}} (\sum_{l=1}^{2} M_{1l} W_{l2}^{R}) \tag{39}$$

$$\frac{\partial L}{\partial W_{12}^{L}} = \frac{\partial L}{\partial M'_{11}} \frac{\partial M'_{11}}{\partial W_{12}^{L}} + \frac{\partial L}{\partial M'_{12}} \frac{\partial M'_{12}}{\partial W_{12}^{L}} = \frac{\partial L}{\partial M'_{11}} (\sum_{l=1}^{2} M_{2l} W_{l1}^{R}) + \frac{\partial L}{\partial M'_{12}} (\sum_{l=1}^{2} M_{2l} W_{l2}^{R}) \tag{40}$$

$$\frac{\partial L}{\partial W_{21}^{L}} = \frac{\partial L}{\partial M'_{21}} \frac{\partial M'_{21}}{\partial W_{21}^{L}} + \frac{\partial L}{\partial M'_{22}} \frac{\partial M'_{22}}{\partial W_{21}^{L}} = \frac{\partial L}{\partial M'_{21}} (\sum_{l=1}^{2} M_{1l} W_{l1}^{R}) + \frac{\partial L}{\partial M'_{22}} (\sum_{l=1}^{2} M_{1l} W_{l2}^{R}) \tag{41}$$

$$\frac{\partial L}{\partial W_{22}^{L}} = \frac{\partial L}{\partial M'_{21}} \frac{\partial M'_{21}}{\partial W_{22}^{L}} + \frac{\partial L}{\partial M'_{22}} \frac{\partial M'_{22}}{\partial W_{22}^{L}} = \frac{\partial L}{\partial M'_{21}} (\sum_{l=1}^{2} M_{2l} W_{l1}^{R}) + \frac{\partial L}{\partial M'_{22}} (\sum_{l=1}^{2} M_{2l} W_{l2}^{R}) \tag{42}$$

Similarly, we can calculate the partial derivative of $L$ with respect to $W_{ij}^{R}$. Moreover, the partial derivative of $L$ with respect to $M_{ij}$ can be calculated as follows,

$$\frac{\partial L}{\partial M_{11}} = \sum_{p=1}^{2} \sum_{q=1}^{2} \frac{\partial L}{\partial M'_{pq}} \frac{\partial M'_{pq}}{\partial M_{11}} = \sum_{p=1}^{2} \sum_{q=1}^{2} \frac{\partial L}{\partial M'_{pq}} (W_{p1}^{L} W_{1q}^{R}) \tag{43}$$

$$\frac{\partial L}{\partial M_{12}} = \sum_{p=1}^{2} \sum_{q=1}^{2} \frac{\partial L}{\partial M'_{pq}} \frac{\partial M'_{pq}}{\partial M_{12}} = \sum_{p=1}^{2} \sum_{q=1}^{2} \frac{\partial L}{\partial M'_{pq}} (W_{p1}^{L} W_{2q}^{R}) \tag{44}$$

$$\frac{\partial L}{\partial M_{21}} = \sum_{p=1}^{2} \sum_{q=1}^{2} \frac{\partial L}{\partial M'_{pq}} \frac{\partial M'_{pq}}{\partial M_{21}} = \sum_{p=1}^{2} \sum_{q=1}^{2} \frac{\partial L}{\partial M'_{pq}} (W_{p2}^{L} W_{1q}^{R}) \tag{45}$$

$$\frac{\partial L}{\partial M_{22}} = \sum_{p=1}^{2} \sum_{q=1}^{2} \frac{\partial L}{\partial M'_{pq}} \frac{\partial M'_{pq}}{\partial M_{22}} = \sum_{p=1}^{2} \sum_{q=1}^{2} \frac{\partial L}{\partial M'_{pq}} (W_{p2}^{L} W_{2q}^{R}) \tag{46}$$

By summarizing the patterns, we can obtain more universal formulas as follows,

$$M'_{ij} = \sum_{k=1}^{L_{in}} \sum_{l=1}^{R_{in}} W_{ik}^L M_{kl} W_{lj}^R \tag{47}$$

where $L_{in}$ and $R_{in}$ denote the number of rows and columns of the matrix representation $M \in \mathbb{R}^{L_{in} \times R_{in}}$, respectively.

Suppose the loss value for the graph property prediction task is $L$ and the output $M' \in \mathbb{R}^{L_{out} \times R_{out}}$. Then, the universal partial derivative of $L$ with respect to $W_{ij}^L$ can be calculated as follows,

$$\frac{\partial L}{\partial W_{ij}^L} = \sum_{q=1}^{R_{out}} \frac{\partial L}{\partial M'_{iq}} \frac{\partial M'_{iq}}{\partial W_{ij}^L} = \sum_{q=1}^{R_{out}} \sum_{l=1}^{R_{in}} \frac{\partial L}{\partial M'_{iq}} M_{il} W_{lq}^R \tag{48}$$

Similarly, the universal partial derivative of $L$ with respect to $W_{ij}^R$ can be calculated as follows,

$$\frac{\partial L}{\partial W_{ij}^R} = \sum_{p=1}^{L_{out}} \frac{\partial L}{\partial M'_{pj}} \frac{\partial M'_{pj}}{\partial W_{ij}^R} = \sum_{p=1}^{L_{out}} \sum_{k=1}^{L_{in}} \frac{\partial L}{\partial M'_{pj}} W_{pk}^L M_{kj} \tag{49}$$

Finally, the universal partial derivative of $L$ with respect to $M_{ij}$ can be calculated as follows,

$$\frac{\partial L}{\partial M_{ij}} = \sum_{p=1}^{L_{out}} \sum_{q=1}^{R_{out}} \frac{\partial L}{\partial M'_{pq}} \frac{\partial M'_{pq}}{\partial M_{ij}} = \sum_{p=1}^{L_{out}} \sum_{q=1}^{R_{out}} \frac{\partial L}{\partial M'_{pq}} (W_{pi}^L W_{jq}^R) \tag{50}$$

At this point, the complete forward and backward propagation processes of MNN have been derived.

## A.2  PSEUDO-CODE AND DETAIL TIME COMPLEXITY OF MATPOOL

---

**Algorithm 1** Pseudo-code of MatPool

**Input:** Graph-structured data $\mathcal{G}(A, X, E)$ with true label $Y$
**Output:** PEM parameters $\phi_l^N$ and $\phi_l^E$, $l = 1, ..., l_g$; MNN parameters $W_l^L$, $W_l^R$ and activation $\phi_l^M$, $l = 1, ..., l_m$; Neural network $NN(:, \theta)$ with parameters $\theta$;
**repeat**
  $Temp = 0$
  **for** $l = 1$ **to** $l_g$ **do**
    $X = \phi_l^N(A_{pem}X + \phi_l^E(E))$
    $Temp = Temp + X$
  **end for**
  $Pool(\mathcal{G}) = Temp^T Temp$
  $Mat(\mathcal{G}) = Pool(\mathcal{G}) \odot M$
  **for** $l = 1$ **to** $l_m$ **do**
    $Mat(\mathcal{G}) = \phi_l^M(W_l^L Mat(\mathcal{G}) W_l^R)$
  **end for**
  $output = NN(Flatten(Mat(\mathcal{G}), \theta))$
  $Loss = CrossEntropyLoss(output, Y)$
  Update $parameters$ by minimizing $Loss$
**until** $Stop\ criteria$ is $true$

---

Algorithm 1 presents the pseudo-code of MatPool. According to the pseudo-code, we analyze the time complexity of MatPool in detail. Suppose the number of nodes, edges, and feature dimensions are $n$, $e$, and $d$ respectively. The time complexity of PEM focuses on the propagating and aggregating process. In this process, $A_{pem}X$ costs $O(n^2d)$, $\phi_l^E(E)$ costs $O(nd^2)$, and the $\phi_l^N$ assumed as a linear mapping costs $O(nd^2)$. In summary, the time complexity of PEM is $O(n^2d + ned + nd^2)$.

The time complexity of MR focuses on the multiplication operation that costs $O(nd^2)$. Moreover, the dot product operation costs $d^2$. Therefore, the time complexity of MR is $O(nd^2)$.

Finally, the time complexity of MNN focuses on the multiplication of two weight matrices. Suppose the output is $\in \mathbb{R}^{d \times d}$, both the left and right multiplications cost $O(d^3)$. Although the time complexity of MNN is $O(d^3)$, it remains fast due to the inherent parallelism of matrix multiplication. Moreover, if the matrix is flattened into a vector, then a linear neural network is used and the time complexity will reach $O(d^4)$.

Therefore, the primary time complexity is concentrated on the propagation and aggregation process of graph neural network module. If the number of GNN layers is $L$, the main time complexity of MatPool is $O(Ln^2d + Lned + Lnd^2)$. This also explains why hierarchical pooling methods consume more training time than global pooling.

## A.3 DETAILED DESCRIPTIONS AND HYPER-PARAMETER SETTINGS OF MATPOOL

Table 5: Summary statistics of datasets.

| Name | Graphs | Avg Nodes | Avg Edges | Classes | Source | Metric |
|---|---|---|---|---|---|---|
| MOLHIV | 41,127 | 25.5 | 27.5 | 2 | OGBG | ROC-AUC |
| MOLPCBA | 437,929 | 26.0 | 28.1 | 2 | OGBG | AP |
| PPA | 158,100 | 243.4 | 2,266.1 | 100 | OGBG | ACC |
| CODE2 | 452,741 | 125.2 | 124.2 | 5002 | OGBG | F1 score |
| AIDS | 2,000 | 15,69 | 16.20 | 2 | Molecules | ACC |
| FRANKENSTEIN | 4,337 | 16.90 | 17.88 | 2 | Molecules | ACC |
| MUTAGENICITY | 4,337 | 30.32 | 30.77 | 2 | Molecules | ACC |
| NCI1 | 4,110 | 29.87 | 32.30 | 2 | Molecules | ACC |
| NCI109 | 4,127 | 29.68 | 32.13 | 2 | Molecules | ACC |
| DD | 1,178 | 284.3 | 715.66 | 2 | Bioinformatics | ACC |
| PROTEINS | 1,113 | 39.06 | 72.82 | 2 | Bioinformatics | ACC |
| COIL-DEL | 3,900 | 21.54 | 54.24 | 100 | Computer Vision | ACC |
| COIL-RAG | 3,900 | 3.01 | 3.02 | 100 | Computer Vision | ACC |
| Letter-high | 2,250 | 4.67 | 4.50 | 15 | Computer Vision | ACC |
| Letter-low | 2,250 | 4.68 | 3.13 | 15 | Computer Vision | ACC |
| Letter-med | 2,250 | 4.67 | 3.21 | 15 | Computer Vision | ACC |
| COLLAB | 5,000 | 74.49 | 2457.78 | 3 | Social Networks | ACC |
| IMDB-BINARY | 1,000 | 19.77 | 96.53 | 2 | Social Networks | ACC |
| IMDB-MULTI | 1,500 | 13.00 | 65.94 | 3 | Social Networks | ACC |
| COLORS-3 | 10,500 | 61.31 | 91.03 | 13 | Synthetic | ACC |

In this work, we select 20 datasets to validate the performance of MatPool and other comparison algorithms. These datasets come from seven categories: OGBG, Molecules, Bioinformatics, Computer Vision, Social Networks, and Synthetic. Detailed descriptions of these datasets are provided in Table 5.

Table 6 lists the detailed descriptions of the hyper-parameters. In the experiments, TUDatasets generally use different hyper-parameters, and there are two tuning hyper-parameters including the learning rate selected from $\{0.001, 0.0001\}$ and the batch size selected from $\{32, 128\}$. The other hyper-parameters are fixed. Accordingly, there are two combinations of hyper-parameters on TUDatasets. We experiment with each combination of hyper-parameters and selected the best combination on the validation set to predict the test set.

## A.4 CONVERGENCE AND HYPER-PARAMETERS OF MATPOOL

**Convergence**: 7 shows the values of losses and the corresponding validation score on the OGBG graph datasets. From the figure, the convergence speed of MatPool on these OGBG graph datasets is relatively fast, and the scores of the validation set can also be rapidly improved in the early stages of training.

**Hyper-parameters**: The detailed setting of hyper-parameter in Table 6. In our method, two hyper-parameters, including learning rate and batch size, are adjusted for different graphs on TUDatasets. The batch size is set to 32 for OGBG-PPA, because a large batch size for OGBG-PPA frequently occurs non-convergence. Therefore, We tune the learning rate on 16 graph datasets from TUDataset.

Table 6: Summary statistics of used hyper-parameters in the experiments.

| Hyper-parameters | MOLHIV/MOLPCBA | PPA | CODE2 | TUDatasets |
|---|---|---|---|---|
| Learning rate | 0.0001 | 0.001 | 0.001 | {0.001, 0.0001} |
| Embedding dim | 256 | 256 | 256 | 256 |
| Batch size | 128 | 32 | 128 | {32, 128} |
| Max epochs | 100 | 100 | 25 | 100 |
| GNN layers | 3 | 3 | 3 | 3 |
| Least epoch | 20 | 20 | 20 | 30 |
| Early stop patient | 15 | 15 | 15 | 20 |
| Learning rate decay | 0.95 | 0.95 | 0.95 | 0.95 |
| Weight decay | 1e-8 | 1e-8 | 1e-8 | 1e-8 |
| Droupout | 0 | 0 | 0 | 0 |
| Run times | 10 | 10 | 10 | 10 |
| Random seeds | 0∼9 | 0∼9 | 0∼9 | 0∼9 |
| Max seq length | NA | NA | 5 | NA |
| Number of vocabulary | NA | NA | 5000 | NA |

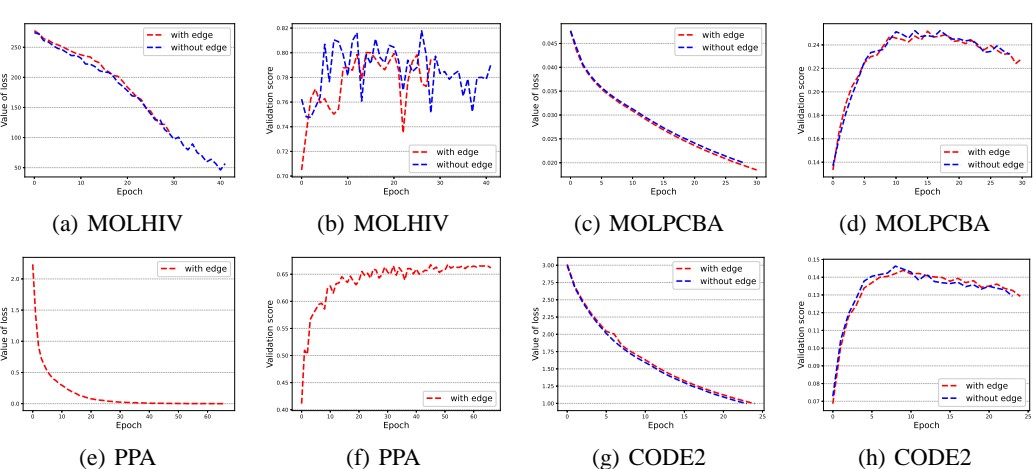

| | | | |
|---|---|---|---|
| (a) MOLHIV | (b) MOLHIV | (c) MOLPCBA | (d) MOLPCBA |
| (e) PPA | (f) PPA | (g) CODE2 | (h) CODE2 |

Figure 7: The figures are corresponding to the values of loss and the validation scores of MatPool on OGBG graph datasets.

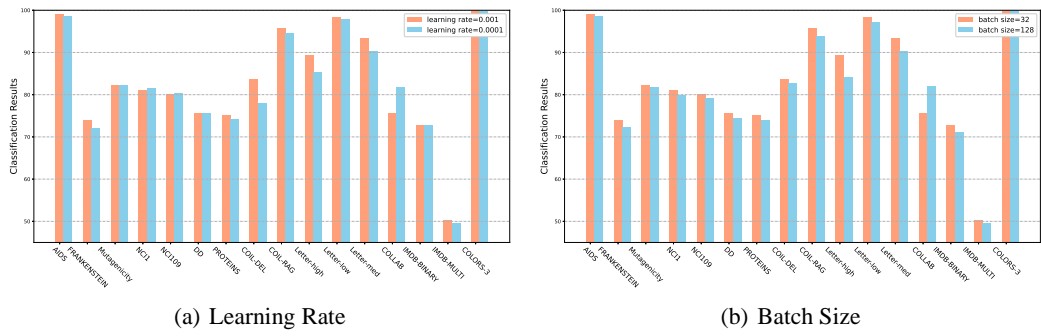

| | |
|---|---|
| (a) Learning Rate | (b) Batch Size |

Figure 8: The sub-figure in the left is the classification result under different learning rate, while that in the right is the classification result under different batch size.

Table 7: Comparison results (%) between MatPool and other classical pooling methods are reported here (The best result on each data set is written in bold).

| Name | MatPool | DKEPool | GMT | MinCutPool | StructPool | DiffPool | SortPool |
|---|---|---|---|---|---|---|---|
| DD | $75.60_{\pm 0.21}$ | $75.26_{\pm 0.47}$ | $\mathbf{78.72}_{\pm 0.59}$ | $78.22_{\pm 0.54}$ | $78.45_{\pm 0.40}$ | $77.56_{\pm 0.41}$ | $75.58_{\pm 0.72}$ |
| PROTEINS | $75.14_{\pm 0.54}$ | $74.37_{\pm 0.50}$ | $75.09_{\pm 0.59}$ | $74.72_{\pm 0.48}$ | $\mathbf{75.16}_{\pm 0.86}$ | $73.03_{\pm 1.00}$ | $73.17_{\pm 0.88}$ |
| MUTAG | $\mathbf{88.33}_{\pm 0.56}$ | $\mathbf{88.33}_{\pm 1.37}$ | $83.44_{\pm 1.33}$ | $79.17_{\pm 1.64}$ | $79.50_{\pm 1.75}$ | $79.22_{\pm 1.02}$ | $71.94_{\pm 3.55}$ |
| HIV | $\mathbf{78.90}_{\pm 0.53}$ | $78.30_{\pm 0.56}$ | $77.56_{\pm 1.25}$ | $75.37_{\pm 2.05}$ | $75.85_{\pm 1.81}$ | $75.64_{\pm 1.86}$ | $71.82_{\pm 1.63}$ |
| Tox21 | $75.93_{\pm 0.10}$ | $75.96_{\pm 0.36}$ | $\mathbf{77.30}_{\pm 0.59}$ | $75.11_{\pm 0.69}$ | $75.43_{\pm 0.79}$ | $74.88_{\pm 0.81}$ | $69.54_{\pm 0.75}$ |
| ToxCast | $\mathbf{65.95}_{\pm 1.03}$ | $64.35_{\pm 0.45}$ | $65.44_{\pm 0.58}$ | $62.48_{\pm 1.33}$ | $62.17_{\pm 1.61}$ | $62.28_{\pm 0.56}$ | $58.69_{\pm 1.71}$ |
| BBBP | $\mathbf{69.47}_{\pm 0.45}$ | $68.10_{\pm 0.79}$ | $68.31_{\pm 1.62}$ | $65.97_{\pm 1.13}$ | $67.01_{\pm 2.65}$ | $68.25_{\pm 0.96}$ | $65.98_{\pm 1.70}$ |
| IMDB-B | $\mathbf{73.75}_{\pm 1.05}$ | $73.05_{\pm 0.95}$ | $73.48_{\pm 0.76}$ | $72.65_{\pm 0.75}$ | $72.06_{\pm 0.64}$ | $73.14_{\pm 0.70}$ | $72.12_{\pm 1.12}$ |
| IMDB-M | $49.47_{\pm 0.53}$ | $51.00_{\pm 0.13}$ | $50.66_{\pm 0.82}$ | $51.04_{\pm 0.70}$ | $50.23_{\pm 0.53}$ | $\mathbf{51.31}_{\pm 0.72}$ | $48.18_{\pm 0.83}$ |
| COLLAB | $\mathbf{82.00}_{\pm 0.30}$ | $81.01_{\pm 0.19}$ | $80.74_{\pm 0.54}$ | $80.87_{\pm 0.34}$ | $77.27_{\pm 0.51}$ | $78.68_{\pm 0.43}$ | $77.87_{\pm 0.47}$ |
| Average | $\mathbf{73.45}_{\pm 0.53}$ | $72.97_{\pm 0.58}$ | $73.07_{\pm 0.87}$ | $71.56_{\pm 0.97}$ | $71.31_{\pm 1.16}$ | $71.40_{\pm 0.85}$ | $68.49_{\pm 1.34}$ |

From Figure 8, the left sub-figure shows that when the learning rate of MatPool is set to 0.001, optimal results are achieved in all datasets except for the COLLAB, NCI1 and, NCI109 datasets. The right figure indicates that when the batch size of MatPool is set to 32, the results are better than when the batch size is set to 128. Therefore, a smaller batch size and a learning rate of 0.001 are generally more suitable for MatPool.

## A.5 COMPARISON WITH OTHER CLASSICAL POOLING METHODS

We compare MatPool with other important baselines such as DiffPool, MuchPool, GMT, StructPool, MinCutPool, DKEPool, and SortPool. In this experiment, the experimental settings of these algorithms are consistent with those of GMT, and the experimental results are directly derived from GMT's results.

From Table 7, MatPool achieved optimal results on 6 out of 10 datasets, and its average results are also the best. Compared to GMT, it performs worse only on the DD and Tox21 datasets, while it outperforms GMT on the remaining datasets. Overall, MatPool also performs better than DKEPool. Compare to the remaining methods, MatPoll has significant advantages. Therefore, we can conclude that MatPool is a simple yet powerful global pooling method.

