# OpenReview forum: "MatPool: Matrix-pattern-oriented Pooling for Graph Property Prediction"
_ICLR.cc/2025/Conference — ICLR 2025 Conference Withdrawn Submission_

### Official Review · Reviewer_6EYb · 2024-11-01

**Soundness:** 2
**Presentation:** 2
**Contribution:** 3
**Rating:** 5
**Confidence:** 3

**Summary:**

The paper proposes Matrix-pattern-oriented Pool (MatPool), a framework for graph representation learning. MatPool leverages Matrix Representation and a Matrix Neural Network to predict graph properties across varying graph sizes, maintaining comprehensive graph information throughout. Experimental results demonstrate the effectiveness of MatPool in diverse graph-based tasks.

**Strengths:**

• The proposed PEM effectively reconstructs the adjacency matrix to ensure positive eigenvalues, which enhances the propagation capacity of primary nodes.
• The Matrix Neural Network (MNN) functions as a global pooling mechanism, offering a straightforward implementation and fast training speed. This approach demonstrates superior performance compared to existing global pooling methods.

**Weaknesses:**

• Certain aspects of MatPool require further clarification. For example, the initialization of the M matrix in Equation 9 is unclear. While it appears to function as a learnable matrix, it is not specified as a learnable parameter in Algorithm 1. Given the importance of matrix M, as demonstrated in Table 3, a more detailed description of its initialization and role in the framework would be beneficial.

**Questions:**

• While graph sizes for graph-level representation tasks are typically small, the Mat(G_N) matrix in MatPool may still demand substantial GPU memory. An analysis of GPU memory usage during training would be helpful for understanding the framework’s computational requirements.

---

### Official Review · Reviewer_V2yt · 2024-11-02

**Soundness:** 2
**Presentation:** 1
**Contribution:** 2
**Rating:** 3
**Confidence:** 3

**Summary:**

The authors propose a message passing approach, which is slightly different than classical GCN denoted PEM, which is basically (and not quite clearly why) adding the sum to the diagonal (opposite to classical Laplacian). They then propose a version of quadratic representation of the graph (again slightly different from current quadratic networks), but the main difference is a multiplication by M as defined in line 265 (between proposition 3.7 and 3.8). The result is as in any other quadratic approach a constant side representation that is then used as the input for classification
The author then present some comparison of multiple methods they proposed. They also compare to other methods in Table 7 in the appendix, but the table show no significant difference between the methods proposed here and previous methods.

**Strengths:**

The method is one more method for presentation of graphs in constant dimensions
The projection is one to one from the graph to the presentation.
There reported performance are as good as the state of the art

**Weaknesses:**

The paper has many of the important details in the appendix, making it de-facto very long. It is impossible to understand without the appendix
The logic of PEM is not very clear
They are far from being the first to propose quadratic methods, but this is not reported.

**Questions:**

An explanation why PEM is better than other methods (or a clearer explanation the idea of putting all the edges in the diagonal) would be more than welcome
It is not clear how is this work better than the existing methods (it is clearly not worse, but does not seem to be better).
The method assumes that the edge feature and vertex feature dimensions are equal. This seems unrealistic. How is this handled in real-world graph datasets?

---

### Official Review · Reviewer_3ty4 · 2024-11-03

**Soundness:** 3
**Presentation:** 3
**Contribution:** 3
**Rating:** 5
**Confidence:** 3

**Summary:**

The article introduces a novel approach to graph property prediction using a matrix-pattern-oriented pooling algorithm, MatPool. Unlike traditional methods that often lose information through graph pooling, MatPool generates a Matrix Representation (MR) by multiplying the feature matrix with its transpose, preserving graph information and ensuring permutation invariance. It employs a Matrix Neural Network (MNN) with two-sided weight matrices to align with row-column correlations. Theoretical analyses support the method's efficacy, and extensive experiments demonstrate its efficiency and effectiveness across various benchmarks, offering a valuable new perspective for graph property prediction.

**Strengths:**

1. The experiments in this paper are comprehensive, conducted across a total of 20 datasets, providing a high level of credibility.
2. The paper includes solid theoretical derivations, enhancing the model's reliability.
3. The paper appears to be easy to follow.

**Weaknesses:**

1. While the Matrix Neural Network (MNN) can enhance computational efficiency through matrix operations, a practical issue arises when multiple graphs are inputted; calculating the Matrix Representation (Mat) may involve redundant computations. (There are computations between different graphs) Are there solutions to address this?
2. The proposed MNN framework can be categorized within the Kernel framework presented in [1], suggesting it functions as a column-based kernel computation. The authors should consider comparing their method with [1] as a baseline.
3. Since M is a learnable matrix, does its initialization method significantly impact the results? Have the authors attempted to incorporate any prior knowledge into the initialization of this matrix?
[1] Yu J, Wu Z, Cai J, et al. Kernel Readout for Graph Neural Networks.

**Questions:**

1. How can the redundancy in computations be minimized when multiple graphs are inputted into the MNN model to calculate the Matrix Representation (Mat)?
2. How does the initialization method of the learnable matrix M affect the results? Have the authors explored incorporating prior knowledge into its initialization?

---

### Official Review · Reviewer_rKYK · 2024-11-06

**Soundness:** 3
**Presentation:** 2
**Contribution:** 2
**Rating:** 5
**Confidence:** 3

**Summary:**

The authors propose MatPool, a novel algorithm for graph property prediction in the context of Graph Neural Networks (GNNs). The method is based on two main components: a new message-passing scheme, called Positive Eigenvalue Mapping (PEM), and a neural network designed to process the resulting matrix-level representation.

First, the authors introduce PEM, a technique that keeps the eigenvalues of the adjacency matrix positive, enhancing the influence of primary nodes and facilitating the aggregation of node features. This process leads to the construction of a graph-level representation, the Matrix Representation (MR), which preserves the structural information of the graph.

Finally, they present the Matrix Neural Network (MNN), designed to extract deeper features from the MR. This new architecture should offer advantages in both execution speed and overall model performance.

**Strengths:**

The construction of the MR is particularly interesting for its ability to preserve information.

**Weaknesses:**

The method is interesting, but the authors should clarify the different contributions and their specific roles starting from the abstract (and in the relevant sections). To be clear, I’m not saying the contributions weren’t listed, but the way and place they were presented could sometimes confuse the reader. For example, PEM isn’t mentioned in the abstract. Moreover, in section 3.2, PEM seems to focus only on aggregation. So, how does pooling fit into this, and how is it defined? It might be helpful to include pooling in equation 9 and also mention it in figure 2. Both sections could be confusing for the reader, so I’d suggest reorganizing sections 3.2 and 3.3 for clarity.

Furthermore, I couldn’t find any explanation for why the method doesn’t perform particularly well on certain datasets, such as Letter-med and Letter-high.

It would have been interesting to consider the use of the normalized Laplacian as well. A comparison with the adjacency matrix, or at least an explanation of why the adjacency matrix was chosen, would have added value to the work.

Another aspect that needs clarification is the model’s scalability: it’s not clear if it can handle very large graphs or networks with millions of nodes and edges.

The paper states that PEM “enhances the influence of primary nodes.” It would be helpful to see experiments (or toy examples) demonstrating the concrete effect of this reinforcement on the model.

Finally, regarding execution speed, I expected a more substantial improvement. Looking at figure 5, the method is almost always on par with other approaches like SOPool or GA.

**Questions:**

It’s unclear whether directed graphs were used in the experiments. If they were not included, it would be helpful to explain why and to add experiments that incorporate them.

The caption for “Table 4: Experimental results (%) for all pooling methods using PEM as the message-passing way are reported here” is somewhat unclear. In section 4.3, it states, “Table 4 shows the experimental results of all pooling methods across the graph datasets used. MatPool achieves the highest results on 11 out of 20 datasets and has the highest average result. Overall, global pooling methods outperform hierarchical pooling methods, and MatPool performs better than other global pooling methods.” This raises the question of how PEM was used; I’d suggest clarifying this point.

How generalizable is the MNN?

---

### Note · Authors · 2025-01-21

I have read and agree with the venue's withdrawal policy on behalf of myself and my co-authors.